# Self-supervised Pitch Detection by Inverse Audio Synthesis

Jesse Engel [1]   Rigel Swavely [1]   Adam Roberts [1]   Lamtharn (Hanoi) Hantrakul [1]   Curtis Hawthorne [1]

## Abstract

Audio scene understanding, parsing sound into a hierarchy of meaningful parts, is an open problem in representation learning. Sound is a particularly challenging domain due to its high dimensionality, sequential dependencies and hierarchical structure. Differentiable Digital Signal Processing (DDSP) greatly simplifies the forward problem of generating audio by introducing differentiable synthesizer and effects modules that combine strong signal priors with end-to-end learning. Here, we focus on the inverse problem, inferring synthesis parameters to approximate an audio scene. We demonstrate that DDSP modules can enable a new approach to self-supervision, generating synthetic audio with differentiable synthesizers and training feature extractor networks to infer the synthesis parameters. By building a hierarchy from sinusoidal to harmonic representations, we show that it possible to use such an inverse modeling approach to disentangle pitch from timbre, an important task in audio scene understanding.

## 1. Introduction

While audio scene analysis is typically associated with source separation (Brown & Cooke, 1994), it also encompasses many sound analysis tasks including pitch detection (Kim et al., 2018; Gfeller et al., 2020), phoneme recognition (Koutras et al., 1999), automatic speech recognition (Coy & Barker, 2007), sound localization (Lyon, 1983), and polyphonic instrument transcription (Hawthorne et al., 2018). Since many sources exhibit harmonic resonance, such as voices and vibrating objects (Smith, 2010), disentangling pitch and timbre is an important step in parsing an audio scene (Moerel et al., 2012; Theunissen & Elie, 2014).

Inverse graphics, where the parameters of a rendering engine

[1]Google Research, Brain Team. Correspondence to: Jesse Engel <jesseengel@google.com>.

*Published at the workshop on Self-supervision in Audio and Speech at the $37^{th}$ International Conference on Machine Learning*, Vienna, Austria. Copyright 2020 by the author(s).

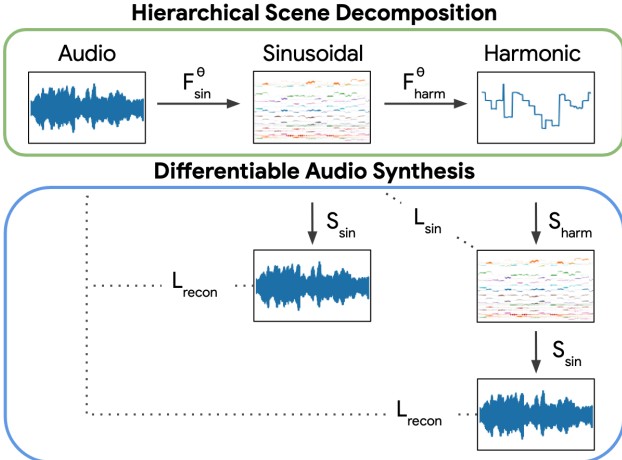

*Figure 1.* Diagram of inverse audio synthesis. A feature extraction pipeline ($F_{sin}^{\theta}$, $F_{harm}^{\theta}$) hierarchically decomposes audio into low-level sinusoidal components (frequency, amplitude), which are combined to extract harmonic components ($f_0$, amplitude, harmonic distribution). These are the only modules that have learnable parameters $\theta$. An additional filtered noise component is not shown. These parameters are fed to differentiable audio synthesizers ($S_{sin}$, $S_{harm}$) and then to reconstruction losses. An additional consistency loss is enforced on the predicted and resynthesized sinusoidal components. See Section 3 for details.

are inferred from an image, is an appealing approach to parsing visual scenes. Unlike black-box classifiers, the approach is object-oriented, interpretable by design, and can generate high-quality images with modern renderers (Wu et al., 2017a; Yao et al., 2018). In audio, these inverse approaches have been limited to the domain of individual sounds from unrealistic commercial synthesizers due to the lack of a realistic, interpretable and differentiable audio rendering engine (Huang et al., 2014; Hoffman & Cook, 2006; Esling et al., 2019).

Most realistic generative models of audio require large autoregressive models that are slow, non-differentiable and cannot generate samples mid-training. (Dieleman et al., 2018; Dhariwal et al., 2020; Hawthorne et al., 2019; Engel et al., 2017; Wang et al., 2017). Differentiable Digital Signal Processing (DDSP) (Engel et al., 2020) overcomes these challenges by combining neural networks with dif-

ferentiable synthesizers to efficiently render realistic audio during training.

Finally, self-supervised techniques typically rely on intrinsic properties of data, such as causality (Oord et al., 2018) or identity-invariance to augmentation (Zhai et al., 2019), to automatically generate supervised labels from datasets. Since our DDSP audio renderer is fully interpretable, we can explore a different form of self-supervision where a fairly generic random process generates both synthetic audio and supervised labels for training. We combine this self-supervision with unsupervised reconstruction losses to adapt to new datasets.

The key contributions of this paper include:

- *DDSP-inv:* An inverse model of sound using DDSP, capable of factorizing pitch and timbre, with comparable pitch detection to SOTA supervised and self-supervised discriminative methods.

- Self-supervised training procedure to train feature extractor networks to infer synthesis parameters from differentiably-rendered synthetic audio.

- *Sinusoidal Synthesizer:* A new DDSP module capable of generating a wide range of audio including inharmonic and polyphonic signals.

- *Sinusoidal Consistency Loss:* A loss function to evaluate the similarity of two arbitrarily-ordered sets of sinusoids and also perform heuristic pitch extraction.

Audio samples are provided in the online supplement at https://goo.gl/magenta/ddsp-inv and code will be available after publication at https://github.com/magenta/ddsp.

## 2. Related Work

**Differentiable Rendering:** Differentiable rendering is a valuable component of inverse graphics models (Loubet et al., 2019; Li et al., 2018b). A natural scene can be "derendered" into a structured object-wise representation via a differentiable shape renderer (Yao et al., 2018) or an explicit scene description that can be recomposed with a graphics engine (Wu et al., 2017b). This literature motivates this work, in which we use DDSP as a differentiable audio renderer.

**Sinusoidal Modeling Synthesis:** The techniques developed by Serra & Smith (1990) model sound as a combination of additive sinusoids and a subtractive filtered noise source. Despite being parametric and using heuristics to infer synthesis parameters, it is a highly expressive model of sound with diverse applications and is even used as a general purpose audio codec in MPEG-4 (Tellman et al., 1995; Klapuri et al., 2000; Purnhagen & Meine, 2000). In

this work, we train neural networks to do this task with end-to-end learning.

**Pitch Detection:** Estimating the fundamental frequency ($f_0$) of a monophonic audio signal, or pitch detection, is a key task to audio scene understanding. We compare against several state-of-the-art baselines in this work. SWIPE (Camacho & Harris, 2008) performs spectrum template matching between the signal and a sawtooth waveform. CREPE (Kim et al., 2018) is a deep convolutional model classifying pitch labels directly from the waveform. SPICE (Gfeller et al., 2020) removes the need for labels by employing self-supervision to predict the frequency shifts applied to training data. While these discriminative methods are trained specifically to detect pitch, DDSP-inv learns to detect $f_0$ as a side-effect of disentangling timbre and pitch in a signal.

## 3. Model Architecture

A diagram and description of our model hierarchy is shown in Figure 1 (*DDSP-inv*, for inverse modeling with DDSP). We describe each component below.

### 3.1. Differentiable Audio Synthesizers

Inspired by the work of Serra & Smith (1990), we model sound as a flexible combination of time-dependent sinusoidal oscillators and filtered noise. From the sinusoids we can infer a corresponding harmonic oscillator with a fundamental frequency. Except for the new sinusoidal synthesizer module, all other modules are identical to the DDSP library introduced in Engel et al. (2020). While other available DDSP modules cover aspects such as room reverberation, we do not consider them here since they are not significant factors in the benchmark datasets.

**Sinusoidal Synthesizer ($S_{sin}$):** We start by creating a new DDSP module that consists of a bank of $K$ sinusoids with individually varying amplitudes $A_k$ and frequencies $f_k$. These are flexibly specified by the output of a neural network $F^\theta_{sin}$ with parameters $\theta$ over $n$ discrete time steps:

$$x(n) = \sum_{k=0}^{K-1} A_k(n)\sin(\phi_k(n)), \qquad (1)$$

where $\phi_k(n)$ is its instantaneous phase obtained by cumulative summation of the instantaneous frequency $f_k(n)$:

$$\phi_k(n) = 2\pi \sum_{m=0}^{n} f_k(m), \qquad (2)$$

The sinusoidal encoder $F^\theta_{sin}$ outputs amplitudes $A_k$ and frequencies $f_k$ every 32ms, which are upsampled to audio rate (16kHz) using overlapping Hann windows and linear interpolation respectively.

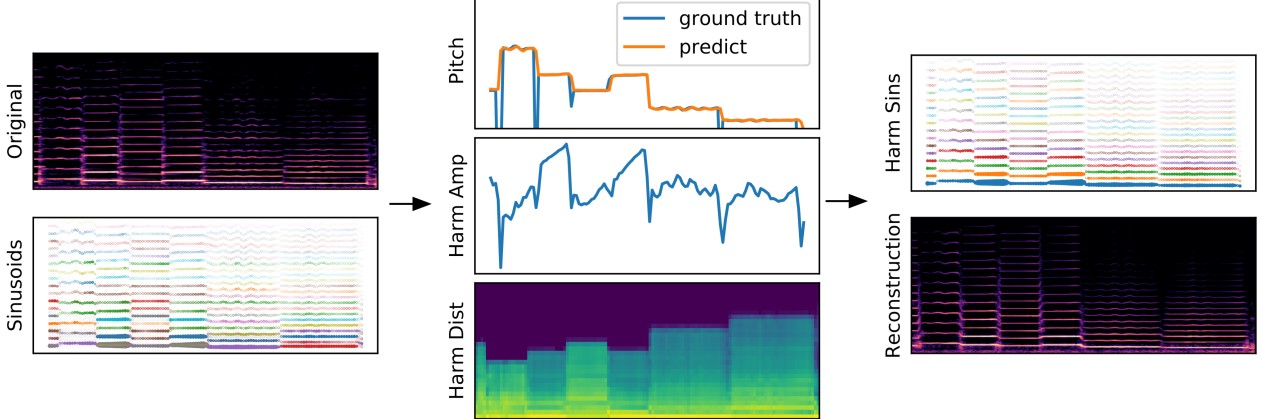

*Figure 2.* Hierarchical decomposition of a sample from the URMP dataset. Left: spectrogram of audio and sinusoidal traces from the sinusoidal encoder $F_{sin}^{\theta}$. Center: harmonic components including fundamental frequency, amplitude, and distribution of the harmonics from the harmonic encoder $F_{harm}^{\theta}$. Right: sinusoids decoded from harmonic components with the harmonic synthesizer $S_{harm}$ and spectrogram of the final reconstructed audio using the sinusoidal synthesizer $S_{sin}$.

We highlight a key difference of this module with a Short Time Fourier Transform (STFT). Frequencies of each sinusoidal component are freely predicted by the network each frame, instead of being locked to a fixed linear spacing determined by the FFT window size. This avoids distortion in periodic signals due to phase mismatch between adjacent frames, and spectral leakage between neighboring frequency bins (Engel et al., 2020).

**Harmonic Synthesizer ($S_{harm}$):** For a harmonic oscillator, the harmonic encoder $F_{harm}^{\theta}$, predicts a single fundamental frequency $f_0$, amplitude $A$, and harmonic distribution $c_k$, from the incoming sinusoids. On generation, all the output frequencies are constrained to be harmonic (integer) multiples of a fundamental frequency (pitch),

$$f_k(n) = k f_0(n) \qquad (3)$$

Individual amplitudes are deterministically retrieved by multiplying the total amplitude, $A(n)$, with the normalized distribution over harmonic amplitudes, $c_k(n)$:

$$A_k(n) = A(n)c_k(n). \qquad (4)$$

where $\sum_{k=0}^{K-1} c_k(n) = 1$ and $c_k(n) \geq 0$.

**Filtered Noise ($S_{noise}$):** As introduced in (Engel et al., 2020), we can model the non-periodic audio components with a linear time-varying filtered noise source. Noise is generated from a uniform distribution. We linearly tile frequency space with 65 bands whose amplitude is modulated each frame by the outputs of the sinusoidal encoder. To ease optimization, we reuse the same filtered noise distribution for both the sinusoidal reconstructions and the harmonic reconstructions.

**Nonlinearities:** For all amplitudes and harmonic distribution components, we constrain network outputs to be positive with a exponentiated sigmoid nonlinearity, $2\sigma(x)^{\log 10} + 10^{-7}$, that scales the output to be between 1e-7 and 2. We constrain sinusoidal frequency predictions between 20Hz and 8000Hz, and harmonic fundamental frequency predictions between 20Hz and 1200Hz. We logarithmically tile 64 bins across this range, then pass network outputs for each frequency component through a softmax nonlinearity across these bins, and take a frequency-bin-weighted sum over the resulting distribution.

### 3.2. Feature Extractors

**Sinusoidal Encoder ($F_{sin}^{\theta}$):** The network converts audio $x(n)$ to sinusoidal amplitudes $A_k$, sinusoidal frequencies $f_k$, and filtered noise magnitudes. Audio is first transformed to a logmel spectrogram (FFT size=2048, hop size=512, mel bins=229), and then passed through a standard implementation of a ResNet-38 with layer normalization, bottleneck layers, and ReLU nonlinearities (He et al., 2016a;b; Ba et al., 2016). Through four stages, the number of filters increases from 64 to 1024, with the frequency dimension downsampling by a factor of two after each stage. A final linear layer feeds the module specific nonlinearities described in Section 3.1.

**Harmonic Encoder ($F_{harm}^{\theta}$):** This network converts the sinusoidal synthesizer components from $F_{sin}^{\theta}$ (amplitudes $A_k(n)$ and frequencies $f_k(n)$ for each sinusoid) into the harmonic synthesizer components of fundamental frequency $f_0(n)$, amplitude $A(n)$, and harmonic distribution $c_k(n)$. Sinusoidal amplitudes and frequencies are first converted to a log scale and fed into a simple network of two fully-

connected layers (256 dims), a single gated-recurrent unit layer (512 dims), and two more fully-connected layers (256 dims). Layer normalization and leaky ReLU nonlinearities are used throughout. A final linear layer feeds the module specific nonlinearities described in Section 3.1.

### 3.3. Loss Functions

We train our network with combination of an audio reconstruction loss, a sinusoidal consistency loss, and a self-supervision loss. We only add the self-supervision loss for synthetic data:

$$\mathcal{L} = \mathcal{L}_{recon} + \alpha_{sin}\mathcal{L}_{sin} + \mathcal{L}_{ss} \quad (5)$$

where $\alpha_{sin}$ is a weight of 0.1 to empirically match the order of magnitude of the other losses.

**Reconstruction Loss:** Since each level of the hierarchical autoencoder can synthesize audio, we can tie the learned representations back to the ground truth audio at each stage with an audio reconstruction loss. Direct waveform comparisons focus too much on absolute phase differences that are less perceptually relevant (Engel et al., 2019). We instead compare spectrograms and utilize the fact that sinusoidal synthesis maintains phase coherence by design. We balance temporal and frequency resolution by imposing a spectrogram loss at several different FFT sizes ($i \in \{64, 128, 256, 512, 1024, 2048\}$) (Wang et al., 2020; Engel et al., 2020):

$$\mathcal{L}_{recon} = \sum_i ||s_i - \hat{s}_i||_1 + ||\log s_i - \log \hat{s}_i||_1. \quad (6)$$

where $s_i$ is the magnitude spectrogram of the target audio at a given FFT size, and $\hat{s}_i$ is the spectrogram of the reconstructed audio. The total reconstruction loss is a sum of the sinusoidal and harmonic reconstruction losses ($\mathcal{L}_{recon} = \mathcal{L}_{recon}^{sin} + \mathcal{L}_{recon}^{harm}$).

**Sinusoidal Consistency Loss:** To compare the *sets* of sinusoids on encoding ($F_{sin}^\theta$) and decoding ($S_{harm}$), we need a permutation invariant loss that can even compare sets of different sizes. We took inspiration from the pitch detection literature; implementing a differentiable version of the Two-Way Mismatch (TWM) algorithm (Maher & Beauchamp, 1994).

The TWM algorithm estimates the distance of two sets of sinusoid frequencies ($f^a$, $f^b$) by the frequency distance from one set to it's closest neighbor in the other set. To prevent the local minima of one set from densely tiling frequency space, the distance is calculated in both directions. This is also called the Chamfer Distance in image recognition literature (Barrow et al., 1977).

$$D_{twm} = \sum_k \min_j(|f_k^a - f_j^b|) + \sum_j \min_k(|f_k^a - f_j^b|) \quad (7)$$

We approximate this procedure as differentiable loss between two arbitrary sets of sinusoids ($A_k^a$, $f_k^a$, $A_j^b$, $f_j^b$), with $K$ and $J$ sinusoids respectively, by creating a Gaussian kernel density estimate (KDE) of $P(f_k^a|A^b, f^b)$ and $P(f_j^b|A^a, f^b)$:

$$p(f_k^a|A^b, f^b) = \sum_j \frac{A_j^b}{\sigma\sqrt{2\pi}} \exp \frac{-(f_k^a - f_j^b)^2}{2\sigma^2} \quad (8)$$

where $A_j$ are the frame-wise normalized amplitudes, and $f_j$ are the component frequencies in units of semitones (logarithmically spaced). The standard deviation of the KDE gaussians, $\sigma$, is a hyperparameter we set to 0.1 semitones.

We then get the loss as a weighted average of the two-way negative log-likelihood:

$$\begin{aligned}\mathcal{L}_{sin} = &-\sum_k A_k^a \log p(f_k^a|A^b, f^b) \\ &-\sum_j A_j^b \log p(f_j^b|A^a, f^a) \\ &+ ||\overline{A^a} - \overline{A^b}||_1\end{aligned} \quad (9)$$

where we use a third term to keep amplitudes bounded by matching their their average value $\overline{A}$ in each frame. An example pair of distributions is shown in Figure 3.

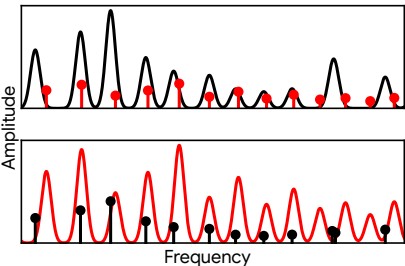

*Figure 3.* Sinusoidal Consistency Loss. Similarity between two sets of sinusoids by a two-way Gaussian kernel density estimate. Stems represent amplitudes of the sinusoids and curves represent the conditional probability distributions of the sinusoids in one plot given the sinusoids in the other plot. TWM loss is minimized when the stems are at the peaks of the Gaussians in both plots.

**TWM Heuristic:** We can also use the sinusoidal consistency loss like the TWM algorithm, as a baseline heuristic for pitch extraction from sinusoids. In this modification, in each time frame we consider all the sinusoids, $f_k$, to be potential candidates as the fundamental frequency, $f_0$, and build a series of harmonics off of each candidate. We then calculate $\mathcal{L}_{sin}$ for each series of harmonics against the original set of sinusoids and take the candidate with the minimum loss.

**Self Supervised Loss on Synthetic Data:** To learn the correct scene decompositions, we found self-supervision with

synthetic data to be an essential addition to reconstruction losses on real data. A bad minimum exists, where fairly good reconstructons are possible by predicting an extremely low fundamental frequency and selectively activating only a few harmonics. This is equivalent to the network learning an STFT representation of the audio, where it chooses the tight linear spacing between frequency bins.

Self-supervised training overcomes this by imposing an implicit prior on the synthesizer parameters. Similar to domain randomization (Tobin et al., 2017), we find diversity of the synthetic data is more important than realism. In our case, we generate notes with variable length, fundamental frequency ($f_0$), amplitude ($A$), harmonic distribution ($c$) and noise magnitudes ($N$). We also add random pitch modulation and noise to all parameters to increase data diversity. Examples are shown in the Supplement Figure 4 alongside further details. The self-supervised loss is given between the true parameters, and those estimated from the synthetic audio (denoted by a hat):

$$\mathcal{L}_{ss} = ||f_0 - \hat{f}_0||_1$$
$$+ \alpha_A||A - \hat{A}||_1$$
$$+ \alpha_c||c - \hat{c}||_1 \qquad (10)$$
$$+ \alpha_N||N - \hat{N}||_1$$
$$+ \alpha_{sin}\mathcal{L}_{sin}(A_k, f_k, \hat{A}_k, \hat{f}_k)$$

where $\alpha_A$, $\alpha_c$, $\alpha_N$, and $\alpha_{sin}$ are loss weights set to 10, 100, 100, and 0.1 to empirically match the order of magnitude of the other losses.

# 4. Experiments

## 4.1. Datasets

We use the following common pitch detection benchmarks in our experiments. We resample all audio to 16kHz, create 4 second long training examples, and randomly partition a 80-20 train-test split.

**MIR-1K:** Hsu & Jang (2009) contains 1,000 clips of people singing Chinese pop songs. Accompaniment music was recorded on the left channel and singing on the right. For our experiments, we used only the singing audio. The dataset includes manual annotations for pitch contours.

**MDB-stem-synth:** Salamon et al. (2017) contains solo recordings of a variety of instruments that were analyzed with pitch tracking techniques and then resynthesized to ensure fully accurate pitch annotations.

**URMP:** Li et al. (2018a) contains recordings of pieces played by small orchestral ensembles. Each instrument for a given piece was recorded in isolation and then later mixed together with the other instruments for the final track. We used only the isolated recordings.

| Raw Pitch Accuracy | MIR-1K | MDB-stem | URMP |
|---|---|---|---|
| *Supervised* | | | |
| SWIPE | 86.6 | 90.7 | - |
| CREPE | 90.1 | **92.7** | **92.2** |
| *Self-Supervised* | | | |
| SPICE | 90.6 | 89.1 | - |
| **DDSP-inv** (this work) | **91.8** | 88.5 | 91.0 |

*Table 1.* Raw pitch detection accuracy. Across a range of instrumental and vocal datasets, DDSP-inv is competitive with SOTA supervised and self-supervised discriminative methods, while also parsing the audio into an interpretable hierarchy of features.

## 4.2. Training Procedure

We find training is more stable by providing a curriculum of first pretraining on synthetic data ($\sim$1M steps) and then fine-tuning training on batches of mixed synthetic and real data ($\sim$100k steps). We use the ADAM optimizer with a batch size of 64 and learning rate of 3e-4, and exponential learning rate decay 0.98 every 10,000 steps (Kingma & Ba, 2015). We also find it helpful to stop direct gradient flow from the harmonic encoder back to the sinusoidal encoder. Note that the two levels are still implicitly connected during training via the sinusoidal consistency loss.

## 4.3. Metrics

We evaluate all models with the standard metrics of Raw Pitch Accuracy (RPA) and Raw Chroma Accuracy (RCA). (Poliner et al., 2007). RPA measures the percentage of voiced frames in which the estimated pitch is within half a semitone of the ground truth pitch. Voiced regions are taken to be frames where the ground truth pitch frequency is greater than 0. RCA is similar to RPA but does not penalize octave errors. The frame is accurate if the predicted pitch is within half a semitone of any power of 2 of the ground truth. Both metrics are computed using the mir_eval python library (Raffel et al., 2014).

## 4.4. Results

Table 1 shows a comparison of SOTA pitch detection methods, both supervised and self-supervised. DDSP-inv outperforms even the supervised models on the singing data of *MIR-1K* and is comparable to other self-supervised methods on the other datasets. Note that while the other models specifically trained to detect pitch, DDSP-inv implicitly learns to detect pitch in a hierarchy of interpretable features.

Table 2 shows the contributions of the harmonic model and real data to model performance. Using the predicted pitch of the harmonic model significantly improves accuracy over the baseline of the Two-way Mismatch (TWM) heuristic on predicted sinusoids. It also dramatically reduces the amount

| RPA (RCA) | MIR-1K | MDB-stem | URMP |
|---|---|---|---|
| *Synthetic Data* | | | |
| TWM | 65.0 (78.6) | 45.6 (75.4) | 50.1 (78.8) |
| DDSP-inv | 77.3 (78.7) | 86.9 (87.1) | 65.3 (69.0) |
| *Synthetic & Real* | | | |
| TWM | 67.2 (86.8) | 60.5 (80.5) | 77.0 (89.7) |
| DDSP-inv | **91.8 (92.0)** | **88.5 (89.6)** | **91.0 (91.8)** |

*Table 2.* Comparison of pitch detection using f0 from the harmonic encoder ($F^{\theta}_{harm}$, DDSP-inv) versus f0 from the sinusoidal encoder ($F^{\theta}_{sin}$) with TWM heuristic. The harmonic model improves accuracy and reduces octave errors, as shown by the reduced gap between RPA and RCA. Real data improves performance, but synthetic data alone is suprisingly effective for some datasets.

of octave errors, as shown by the reduced gap between RPA and RCA. While adding real data makes performance competitive with SOTA, the model achieves fairly good accuracy with synthetic data alone, especially on the *MDB-stem-synth* dataset.

## 5. Conclusion and Future Work

We have presented an interpretable hierarchical model of audio that disentangles timbre and pitch through self-supervised inversion of audio synthesis. We believe this forms a promising foundation for learning higher levels of structure, such as discrete tokens, and extensions to more complicated audio scenes, including polyphonic audio with multiple sources.

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

# Supplement

## 5.1. Synthetic Data

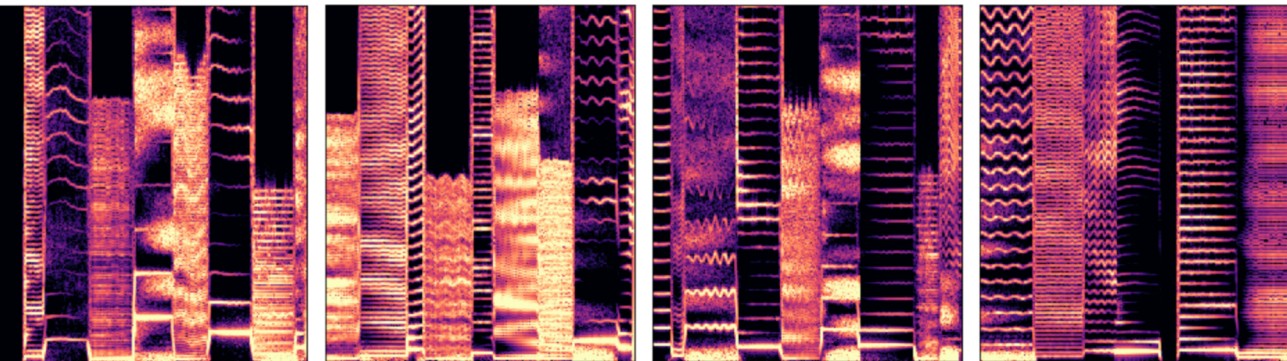

*Figure 4.* Example spectrograms of synthetic data. Notes are first given random lengths and fundamental frequency, with a possibility of being silent. Notes are then given a random amplitude, harmonic distribution, noise distribution at their start and end, and interpolated between. Additional vibrato and parameter noise is then added. Parameters were tuned until the authors subjectively felt that it produced a cool diversity of sounds, even if not particularly realistic. Exact details can be found in the code at `https://github.com/magenta/ddsp`.

---

**Algorithm 1** Generate Synthetic Example

---

```
t <- random note length
With probability p:
  Return silence for length t
Else:
  A_start, A_end <- random harmonic amplitudes
  A <- interpolate(A_start, A_end, t) + noise

  c_start, c_end <- random harmonic distributions
  c <- interpolate(c_start, c_end, t) + noise

  f_0 <- random frequency + random vibrato + noise

  n_start, n_end <- random noise distributions
  n <- interpolate(n_start, n_end, t) + noise

  Return A, c, f_0, n
```

---

## 5.2. Connection of TWM to Jefferys Divergence

It's interesting to note that the sinusoidal consistency loss corresponds to a Jefferys Divergence (Jeffreys, 1946) between two Gaussian KDE distributions $(p, q)$:

$$
\begin{aligned}
D_J &= \frac{1}{2} D_{KL}(p \parallel q) + \frac{1}{2} D_{KL}(q \parallel p) \\
&= -\frac{1}{2} \Big[ \mathop{\mathbb{E}}_{f_k^a \sim p(f_k^a | A^a)} \log p(f_k^a | A^b, f^b) + \mathop{\mathbb{E}}_{f_j^b \sim p(f_j^b | A^b)} \log p(f_j^b | A^a, f^a) \Big]
\end{aligned}
$$
(11)

which is equivalent to Equation 9 (except a factor of 1/2) in the limit that frequencies $f_k$ are sampled proportionally to their normalized amplitudes $A_k$.