# OpenReview forum: "Self-supervised Pitch Detection by Inverse Audio Synthesis"
_ICML.cc/2020/Workshop/SAS — SAS 2020_

### Official Review · AnonReviewer2 · 2020-06-29
**interesting use of Differentiable Digital Signal Processing and self-supervision**

**Rating:** 8
**Confidence:** 3

**Review:**

The paper investigates inverse audio modeling and Differentiable Digital Signal Processing for deriving the synthesis parameters associated to an audio scene. Building on DDSP modules and self-supervision with synthetic data, the authors show how to obtain effective pitch detection in a synthesis framework where audio is first decomposed in sinusoidal and harmonic components used by the differentiable synthesizers.
Overall the paper is well written and the general concept is clearly introduced; moreover, the authors will release the code after publication so experimental results can be reproduced and verified. I suggest to partially expand the section discussing the filtered noise component, where the noise source is not explained, and slightly rearrange the presentation of the proposed losses, since the self-supervised loss is defined in a different section.
Minor points that deserve some details or comments are:
- the weights associated to the sinusoidal consistency loss and the self-supervised loss;
- formula (8) where the self-supervised loss apparently seems to include also the sinusoidal consistency loss;
- in Table 1,for dataset MIR-1K, the self-supervised methods perform better than the supervised methods;
- the definition of harmonic pitch detection (Table 2) is not clearly defined.

---

### Official Review · AnonReviewer1 · 2020-06-29
**Clear, descriptive writing on a powerful line of research**

**Confidence:** 5
**Rating:** 9

**Review:**

This paper describes a framework and methodology for pitch detection by inverse audio synthesis. The broad description of the approach is clear, and the results succinct. Most of my questions then relate to robustness / practical complexity of implementation, as the overarching view of the applicability of invertible and differentiable signal processing modules is one I agree with.

There are a number of custom settings, losses, and nonlinearities in this model. For practical tuning, what were the most critical / touchy aspects of the model? In general, how were things like loss weights, sigma in the TWM, FFT width / step, and the general loss weights of "... weights set to 10, 100,100, and 0.1 respectively to make the losses the same order
of magnitude" (I assume this was analyzed empirically,  then set? or was it derived from the losses construction).

Noting things like "We also find it helpful to stop direct gradient flow from the harmonic encoder back to the sinusoidal encoder" leads me to believe the tuning / setup is not easy, and knowing some of the common pitfalls could speed followup work (thank you for the future code release noted in the paper as well!)

If the FFT width / step rate was radically changed, do you think the same setup would largely work - or would there need to be a decent effort retuning a number of these follow on losses and hyperparameters. Do you have any advice which could aid people applying this general framework to their own data?

Outside the context of this paper, it would be nice to see demonstrations of reconstructions on other, less "harmonic/sinusoidal" sounds (the violin sample is quite nice, but violin is also well suited for this kind of sinusoidal deconstruction / reconstruction). Though this particular paper is describing the use of these tools for pitch detection, it seems the larger mechanisms and framework are generally applicable to synthesis for a host of sounds - such as drums, cymbals, industrial machines, or who knows what else.

Given the large variability of the synthetic training data, it would not be shocking to me that other sounds which were historically *very* difficult to synthesize believably, could be generated in a much more holistic way, and I would be excited to see it. Many of the mechanisms here also closely resemble approaches for voice synthesis / compression (such as WORLD, STRAIGHT, and so on) and may be applicable in modern TTS frameworks as well (as is shown by some of the papers you have cited).

Interesting work, and a well written paper.

---

### Official Review · AnonReviewer3 · 2020-06-30
**An nice application of a recently (re-)discovered technique**

**Confidence:** 3
**Rating:** 8

**Review:**

The paper shows that a recently proposed framework for audio synthesis using (essentially) addition of waveforms of various frequencies can be used to learn a pitch detector by analyzing and re-synthesizing input audio. Pitch corresponds to the main component and is thus easily extracted, if the input audio is passed through the analysis part of the auto-encoder. This approach does not depend on fixed frequencies but works with variable frequencies, which makes the approach accurate and flexible. Many of the technical details are discussed in two recently published (ICLR 2019, ICLR 2020) papers.

The paper is mostly well written, but (mostly) the introduction and related work section needs more work:
- The introduction reads a bit like the related work section already - maybe rework the intro to situate the work in context better, and move some of the references to the related work
- Please cite papers "properly", as conference papers, not just by pointing to their arXiv/ open review IDs
- Please define terms: not all readers may be familiar with "pitch", "timbre" (or "fundamental frequency") and similar terms with very specific meaning (and: https://www.musicradar.com/how-to/understanding-the-difference-between-pitch-and-frequency)
- what does "parsing an audio scene" mean? does (Virtanen, 2007) "parse" audio into a hierarchical representation?
- Please be consistent with citations: how is (Deng, 2013) a good reference for automatic speech recognition? it is not the first paper to do that, it does not discuss closely related work? "identity invariance of augmentation" was only discovered in 2019?

Overall, the paper gives off the air of an application of end-to-end deep learning to an interesting problem, which however deserves to be described with a bit more care and detail.

Strengths:
- an interesting study based on recently released papers (and available software, AFAICT)
- good motivation of the technique
- strong results, novelty appropriate for a workshop paper
- paper overall well written

Weaknesses:
- paper has language issues (but authors should be able to fix these, as they seem to have done in the past)
- references and related work should be done more carefully
- authors could provide more analysis, why does for example their technique not work (relatively) as well for re-synthesised data?

---

### Author Response · Authors · 2020-07-14
**Revisions**

Thank you very much to all the reviewers. We've found your comments very helpful and appreciate the thought and care that went into them. We have done our best to address them all, especially properly handling of references and clarifying the mathematical notation where possible. (We still left the arxiv citation of Layer Norm, because we couldn't find another reference)

---

### Decision · Program_Chairs · 2020-07-01

**Decision:**

Accept

**Comment:**

Dear author(s),

Thank you very much for your submission at the ICML2020@SaS workshop (https://icml-sas.gitlab.io/). Based on the scores assigned by the reviewers, we are happy to notify you that your paper was accepted for the workshop.

Please, address the comments of the reviewers and submit the camera-ready version by July 8. We ask the authors to record a 15min video for your talk. At the workshop, we will have the pre-recorded video as well as a live QA session. It is important to keep this time limit, otherwise, your talk will be automatically cut. The deadline for uploading the video is July 8. The detailed instructions for uploading will follow.

Feel free to contact us for any questions!

Best,

The ICML20@SaS organizers:
Mirco Ravanelli
Titouan Parcollet
Dmitriy Serdyuk
Devon Hjelm
Bhuvana Ramabhadran